# Long-COVID-19 in Asymptomatic, Non-Hospitalized, and Hospitalized Populations: A Cross-Sectional Study

**DOI:** 10.3390/jcm12072613

**Published:** 2023-03-30

**Authors:** Aysegul Bostanci, Umut Gazi, Ozgur Tosun, Kaya Suer, Emine Unal Evren, Hakan Evren, Tamer Sanlidag

**Affiliations:** 1Department of Medical Microbiology and Clinical Microbiology, Faculty of Medicine, Near East University, Nicosia 99138, Cyprus; 2Department of Biostatistics, Faculty of Medicine, Near East University, Nicosia 99138, Cyprus; 3Department of Infectious Diseases and Clinical Microbiology, Faculty of Medicine, Near East University, Nicosia 99138, Cyprus; 4Department of Infectious Diseases and Clinical Microbiology, Faculty of Medicine, University of Kyrenia, Kyrenia 99320, Cyprus; 5DESAM Research Institute, Near East University, Nicosia 99138, Cyprus

**Keywords:** long COVID, questionnaire, asymptomatic, symptomatic, risk factors, Post-COVID-19 syndrome, cross-sectional study

## Abstract

A substantial proportion of coronavirus disease 2019 (COVID-19) survivors continue to suffer from long-COVID-19 (LC) symptoms. Our study aimed to determine the risk factors for LC by using a patient population from Northern Cyprus. Subjects who were diagnosed with severe acute respiratory syndrome-2 (SARS-CoV-2) infection in our university hospital were invited and asked to fill in an online questionnaire. Data from 296 survivors who had recovered from COVID-19 infection at least 28 days prior the study was used in the statistical analysis. For determination of risk factors for “ongoing symptomatic COVID-19 (OSC)” and “Post-COVID-19 (PSC)” syndromes, the patient population was further divided into group 1 (Gr1) and group 2 (Gr2), that included survivors who were diagnosed with COVID-19 within 4-12 weeks and at least three months prior the study, respectively. The number of people with post-vaccination SARS-CoV-2 infection was 266 (89.9%). B.1.617.2 (Delta) (41.9%) was the most common SARS-CoV-2 variant responsible for the infections, followed by BA.1 (Omicron) (34.8%), B.1.1.7 (Alpha) (15.5%), and wild-type SARS-CoV-2 (7.8%). One-hundred-and-nineteen volunteers (40.2%) stated an increased frequency of COVID-19-related symptoms and experienced the symptoms in the week prior to the study. Of those, 81 (38.8%) and 38 (43.7%) were from Gr1 and Gr2 groups, respectively. Female gender, chronic illness, and symptomatic status at PCR testing were identified as risk factors for developing OSC syndrome, while only the latter showed a similar association with PSC symptoms. Our results also suggested that ongoing and persistent COVID-19-related symptoms are not influenced by the initial viral cycle threshold (Ct) values of the SARS-CoV-2, SARS-CoV-2 variant as well as vaccination status and type prior to COVID-19. Therefore, strategies other than vaccination are needed to combat the long-term effect of COVID-19, especially after symptomatic SARS-CoV-2 infection, and their possible economic burden on healthcare settings.

## 1. Introduction

Since its emergence in late 2019, coronavirus disease 2019 (COVID-19) has caused serious health problems worldwide. While the current rates of severe disease and hospitalization are on the decline due to the effective vaccination regimens [1], a substantial proportion of survivors continue to suffer from long-COVID-19 (LC) symptoms for weeks or months after the onset of COVID-19 [2,3,4,5,6]. Today, as the number of people infected with SARS-CoV-2 continues to rise, LC is heralded as the next threat to healthcare systems, which were already overwhelmed during the ongoing COVID-19 pandemic; recently, the Office for National Statistics estimated that almost two million people are experiencing LC symptoms in the United Kingdom [7].

A wide variety of symptoms, such as fatigue, malaise, shortness of breath, cough, and cognitive impairments, can occur in COVID-19 survivors. The prevalence of LC is still not known; due to the differences in study designs including follow-up, the definition of the disease, and region, a broad prevalence range of 22% to 81% was estimated by previous reports [8]. However, a recent review that analyzed more than 190 reports published until January 2022 revealed that at least 45% of COVID-19 survivors experience one and more unresolved symptoms at four months after the onset of SARS-CoV-2 infection [9].

Moreover, the underlying mechanisms responsible for LC are still not well established since LC symptoms may not be specific to COVID-19 and can be associated with post-intensive care syndrome or an exacerbation of pre-existing health conditions. Nevertheless, both the organ damage from the acute infection phase and specific long-lasting inflammatory mechanisms are thought to be involved in the pathophysiology [2]. On the other hand, while the literature on the risk factors is not yet clear on the association of LC with the severity of COVID-19 infection, it consistently reported higher incidence rates in subjects with female gender, old age, and comorbidities [8].

Literature on LC is also difficult to interpret because of variable terms (such as post-acute COVID-19 syndrome or post-COVID conditions) used to define the condition. In an attempt to standardize the terms, the National Institute for Health and Care Excellence (NICE) proposed the use of the term “Long-COVID-19”. The term also covered the “ongoing symptomatic COVID-19 (OSC)” and “Post-COVID-19 syndrome (PSC)”, which are defined as the persistence of symptoms for periods between 4 and 12 weeks, and beyond 12 weeks from the onset of COVID-19, respectively, without any alternative diagnosis [10,11]. Our study aimed to identify the associated risk factors and prevalence of both OSC and PSC in Northern Cyprus, which is yet to receive attention in the literature. For this purpose, COVID-19 survivors who were tested with reverse-transcriptase polymerase chain reaction (RT-PCR) for SARS-CoV-2 at a university hospital were invited to join our study, and then asked to complete an online questionnaire. 

## 2. Materials and Methods

### 2.1. Study Design

This retrospective cohort study was conducted with COVID-19 survivors in Northern Cyprus. Subjects who were previously diagnosed with SARS-CoV-2 infection at the Near East University Hospital COVID-19 PCR Diagnosis Laboratory were reached by phone and invited to participate in an online survey developed on Google Forms. Only data from those who were diagnosed with COVID-19 at least 28 days prior to study was included. Duplicate database entries with the same user ID were eliminated before analysis. Data collected from a total of 296 volunteers between September 2021 and February 2022 was used in a statistical analysis to determine the risk factors associated with LC among the population studied. Information on SARS-CoV-2 variants, RT-PCR Ct values, and vaccination status of the participants were obtained from the hospital database.

### 2.2. Detection of SARS-CoV-2 and its Variants

SARS-CoV-2 infection was diagnosed by RT-PCR performed on nasopharyngeal swab samples, utilizing Uniplex RT-qPCR SARS-CoV-2 RT-qPCR Detection Kit (IKAS Medical, Nicosia, Northern Cyprus) that is based on amplification of viral ORF1ab, N1, and N2 genes and uses human Rnase P as an internal control. SARS-CoV-2 variant analysis was conducted by using Multiplex SARS-CoV-2 VoC RT-qPCR Detection Kit (IKAS Medical, Nicosia, Northern Cyprus) that identifies variants of concern, including B.1.1.7 (Alpha), B.1.351 (Beta), B.1.617.2 (Delta), P.1 (Gamma), and B.1.1.529 (Omicron) variants, by simultaneously detecting mutations (del69/70, N501Y, K417N, T478K, Y144del, and P681R) in the Spike protein gene [12,13]. 

### 2.3. Ethics

The study was conducted in line with the guidelines of the Declaration of Helsinki, and was approved by the ethics committee of the Institutional Review Board at Near East University (YDU/2021-92-1359). Written informed consent was obtained from all participants prior to study enrolment.

### 2.4. Survey

The survey (Appendix A) was comprised of three sections. The first section focused on the sociodemographic characteristics, including age, gender, pre-existing medical comorbidities, and the vaccination status of the participants. The second section included questions on acute symptoms, disease severity, hospitalization, and admission to the intensive care unit (ICU). The third part focused on health status after COVID-19 and LC symptoms.

### 2.5. Statistical Analysis 

Descriptive statistics for the qualitative variables were provided as frequencies and percentages, while the arithmetic mean, standard deviation, median, minimum, and maximum values were calculated for the quantitative variables. The factors that might be associated with post-COVID symptoms were tested using the Pearson chi-square test or Fisher’s exact test, where appropriate. Odds ratios with a 95% confidence interval were calculated. Binary logistic regression analysis was performed to calculate the odds ratios and significance for ordinal qualitative risk factors with more than 2 categories. The level of significance was accepted as 0.05. All statistical analyses were performed with SPSS (Version 26.0 for Mac) software.

## 3. Results

### 3.1. Characteristics of Participants

A total of 296 participants, with an average age of 37.2 ± 14.9 years (range: 12–83 years), were included in the study. The average time interval between the onset of SARS-CoV-2 infection and filling the questionnaire was 3.4 months (range: 1–23 months, ± 3.5).

The numbers of COVID-19 survivors aged ≤17, 18–55, and ≥56 years were 23 (7.8%), 233 (78.7%), and 40 (13.5%), respectively. Of the participants, 149 (50.3%) were male, and 147 (49.7%) were female. A total of 81 (27.4%) participants have at least one chronic disease (comorbidity). The numbers of smokers and non-smokers were 67 (22.6%) and 227 (76.7%), respectively, while no relevant data on smoking habit was obtained from the two subjects (Table 1). 

Two-hundred-and-sixty-six participants (89.9%) were vaccinated before SARS-CoV-2 infection. Of those, while 191 (71.8%) completed vaccination regiments with Coronavac (n = 58; 30.4%), Pfizer (n = 94; 49.2%), Moderna (n = 3; 1.6%), Johnson & Johnson (n = 28; 14.7%), and Oxford-Astra Zeneca (n = 8; 4.2%), the remaining (n = 75, 28.2%) took one dose of the Coronavac (n = 9; 12.0%), Pfizer (n = 65; 86.7%), and Oxford-Astra Zeneca (n = 1; 1.3%) vaccine. For our analysis, data from only those who completed the vaccination regiments were used, and in order to increase the sample size for higher statistical power, the groups were merged according to the type of vaccine received; killed-virus (Coronavac), mRNA (Pfizer +Moderna), and vector-based (Johnson +Astra Zeneca) (Table 1). 

Among the COVID-19 survivors, 254 (85.8%) declared that they were symptomatic during the RT-PCR detectable phase. Twenty-nine of the participants (9.8%) were hospitalized and eight (27.6%) required admission to the ICU. According to the variant analysis, the numbers of B.1.617.2 (Delta), BA.1 (Omicron), B.1.1.7 (Alpha) variants, and wild-type cases SARS-CoV-2 were 124 (41.9%), 103 (34.8%), 46 (15.5%), and 23 (7.8%), respectively (Table 1). 

### 3.2. Risk Factors Associated with OSC and PSC

According to the survey results, 136 volunteers (45.9%) experienced an increased frequency of COVID-19-related symptoms, such as fatigue (n = 56, 41.1%), cough (n = 35, 25.7%), memory problems (n = 28, 20.6%), dyspnea (n = 26, 19.1%), and headache (n = 22, 16.2%), after recovering from SARS-CoV-2 infection. The number of subjects with ongoing LC symptoms (i.e., volunteers who experienced at least one LC symptom in the week prior to the study) was 119 (40.2%) (Table 2). 

For determination of risk factors associated with OSC and PSC, the COVID-19 survivors were divided into two groups depending on the time since COVID-19 diagnosis; while group 1 (Gr1) included survivors, who were diagnosed with SARS-CoV-2 infection from 4 to 12 weeks before the study conducted, Gr2 included participants who joined the study at least three months after the onset of infection. 

The prevalence of the LC symptoms in the Gr1 and Gr2 groups were 66.2% and 33.8%, respectively. The most common symptoms in Gr1 subjects were fatigue and cough, while they were fatigue and dyspnea in Gr2 members. Of Gr1 subjects, the number of patients with ongoing LC symptoms was 81 (38.8%), while the corresponding number was 38 (43.7%) for Gr2 volunteers. The most common ongoing symptoms reported by Gr1 and Gr2 volunteers were fatigue (n = 34, 37.7% for Gr1; n = 22, 47.8% for Gr2), cough (n = 27, 30.0% for Gr1; n = 8, 17.4% for Gr2), memory problems (n = 22, 24.4% for Gr1; n = 6, 13.0% for Gr2), dyspnea (n = 16, 17.8% for Gr1; n = 10, 21.7% for Gr2), and headache (n = 15, 16.7% for Gr1; n = 7, 15.2% for Gr2) (Table 2). 

The associations of OSC and PSC with different risk factors were evaluated by using data provided by subjects with ongoing LC symptoms (81 Gr1 and 38 Gr2 volunteers). Among the different risk factors evaluated, female gender (*p* = 0.006), presence of chronic disease (*p* = 0.007), and symptomatic status at PCR testing (*p* = 0.001) displayed a statically significant association with the incidence of persistent COVID-19 symptoms in Gr1 (Table 3). The incidence rate of OSC was 2.3 higher in female than in male participants, while subjects with chronic disease and COVID-19 symptoms at PCR testing displayed a 2.4- and 14.8-fold higher risk for OSC, respectively, than those without (Table 3). In contrast, among the risk factors associated with OSC, only symptomatic status at PCR testing (*p* = 0.015) showed an association with PSC; volunteers with symptoms exhibited a 9.0 higher incidence rate of persistent symptoms (Table 4). Nevertheless, data on SARS-CoV-2 variants and vaccination regimens could not be used to evaluate their correlation with LC sub-groups due to restrictions imposed by low sample size. However, our statistical analysis revealed that neither of the variables was associated with LC (Table 5). 

## 4. Discussion

The SARS-CoV-2 infection is characterized by a wide spectrum of clinical profiles ranging from asymptomatic to severe COVID-19 disease associated with acute respiratory distress syndrome (ARDS), that can lead to morbidity and mortality from alveolar lumen damage. Today, the risk of becoming severely ill from COVID-19 is significantly lower than that seen in prior estimates because of protection provided by vaccination against SARS-CoV-2. Nevertheless, LC, which is defined by the persistence of COVID-19-related symptoms for weeks and months after the onset of infection with SARS-CoV-2, is predicted to be the next global health crisis with the growing burden on healthcare systems [14,15]. Subjects with persistent LC symptoms may have difficulty to perform daily activities and return to work that can negatively impact their quality of life and lead to great social as well as economic consequences [16]. Our study aimed to evaluate the health status of COVID-19 survivors and determine the risk factors associated with OSC and PSC in Northern Cyprus. The results can provide valuable information for policymakers to develop strategies to combat against long-term effects of SARS-CoV-2 infection.

According to our results, the prevalence of LC among COVID-19 survivors in Northern Cyprus is more than 45%, which is within the range (22–81%) obtained from previous studies [8]. When the prevalence was further analyzed for LC subtypes, 66.2% and 33.8% of participants were found to experience OSC and PSC, respectively. In correlation with previous studies, the most prevalent LC symptoms reported in our study were fatigue and cough [4,5,17]. On the other hand, while the most common symptoms in subjects with OSC syndrome were fatigue and cough, they were fatigue and dyspnea in PSC patients, which is in correlation with previous findings [4,18,19,20].

The statistical analysis revealed that female gender, chronic disease, and symptomatic status at PCR testing are risk factors associated with OSC, while only the latter exhibited a correlation with PSC [5,11,21]. Female gender and the presence of a comorbidity did not have any influence on the rate of PSC syndrome, which was in contrast to previous findings [20]. Moreover, our study reported a lack of association of OSC and PSC with Ct values detected in the acute phase of infection, which contradicts with the data presented by Perez et al. showing a negative correlation between the viral load and the number of the LC symptoms [22]. Additionally, age, which was also inversely correlated with LC symptoms in a recent report [23], was found to be a significant risk factor for neither OSC nor PSC in our study. While the conflict between our results and previous findings can be because of differences in the methodology and populations used by the studies, it can also be due to the underlying bias related to the self-reported nature of our data.

Increasing the COVID-19 vaccination rate is effective in reducing severe disease and hospitalization; however, it does not influence post-COVID-19 recovery since being unvaccinated was not a risk factor for developing either OSC or PSC in our study. Moreover, the type of vaccine received did not have any effect on the development of LC. Accordingly, in a recent systematic review and meta-analysis, the protective effect of COVID-19 vaccines was suggested for some of the LC symptoms, such as cognitive dysfunction/symptoms, kidney diseases/problems, myalgia, and sleeping disorders/problems, while it was not evident for others, including chest/throat pain, fatigue, headache, and respiratory symptoms [24]. Therefore, the potential protective effect of vaccination against specific OSC and PCS symptoms needs further clarification from future studies. On the other hand, this lack of effect highlights the importance of strategies other than promoting vaccination to combat against the long-term effects of SARS-CoV-2 infection. One such strategy could involve the introduction of a remote patient monitoring (RPM) program that enables the patients to transmit health data at home by using phone calls or telemonitoring applications [25].

While the emergence of new SARS-CoV-2 variants was initially thought to influence LC rates, the previous studies have failed to report any association [26,27]. However, to the best of our knowledge, there has not been any relevant study simultaneously comparing the frequencies of LC syndrome between subjects infected with wild-type, B.1.1.7 (Alpha), B.1.617.2 (Delta), and BA.1 (Omicron) variants of SARS-CoV-2. This was addressed by our study, which demonstrated similar percentages of LC between the volunteers exposed to either SARS-CoV-2 variant. However, due to the small sample size, it was not possible to evaluate their association with OSC and PSC separately. Therefore, studies with bigger sample sizes are required to investigate their potential difference in their ability to cause OSC and PSC.

In our analysis, Delta and Omicron were reported to be the two most common SARS-CoV-2 variants; they were responsible for >75% of infections in our study population, most of whom (>95.0%) tested positive for COVID-19 between January 2021 and February 2022. This is in correlation with literature suggesting Delta and Omicron as the two dominant SARS-CoV-2 variants in 2021 [28]. On the other hand, according to our hospital database, none of the volunteers were infected with the Beta variant, which could be due to its low prevalence during the same period [29].

Apart from the self-reported nature of the presented data that may lead to an over-estimation of LC prevalence, the other weaknesses of our study are that the participants were not evenly distributed among groups, and the majority (>90%) of the participants were non-hospitalized patients. Moreover, our study did not include a control group; since ongoing/persistent COVID-19-related symptoms are common and can also be caused by other microbial infections, inclusion of a control group would have helped us to discriminate between the symptoms of those with and without SARS-CoV-2 exposure. Therefore, the results presented in this study should be interpreted with caution. Future studies with the inclusion of bigger sample sizes, physiological assessment/clinical examinations, and controlled or baseline comparison groups are of vital importance to confirm our data.

## 5. Conclusions

Our findings reveal that more than 45% of COVID-19 survivors in Northern Cyprus experience LC symptoms, while the prevalence of OSC and PSC were more than 60% and 30%, respectively. According to our analysis, COVID-19 survivors with female gender, chronic disease, and symptoms at PCR testing are susceptible to suffering from OSC, while only the latter factor was associated with PSC. Furthermore, the results show a lack of association of vaccination status, SARS-CoV-2 variants, and viral load in the acute phase of SARS-CoV-2 infection with ongoing and persistent COVID-19 symptoms. Therefore, strategies other than promoting vaccination are required to combat against the long-term effects of COVID-19, especially after symptomatic SARS-CoV-2 infection.

## Figures and Tables

**Table 1 jcm-12-02613-t001:** Demographic characteristics and clinical profile of the participants.

Characteristics		Gr1 n (%)	Gr2 n (%)	Total n (%)
**Gender**	Female	102 (48.8)	45 (51.7)	147 (100.0)
	Male	107 (51.2)	42 (48.3)	149 (100.0)
**Age**	12–17 years	18 (8.6)	5 (5.7)	23 (100.0)
	18–55 years	170 (81.3)	63 (72.4)	233 (100.0)
	(56 years	21 (10.0)	19 (21.8)	40 (100.0)
**Smoking status**	Smoker	47 (22.5)	20 (23.0)	67 (100.0)
	Non-smoker	147 (70.3)	62 (71.3)	209 (100.0)
	Former-smoker	13 (6.2)	5 (5.7)	18 (100.0)
**Chronic disease**	Present	55 (26.3)	26 (29.9)	81 (100.0)
	Absent	154 (73.7)	61 (70.1)	215 (100.0)
**Vaccination status**	Vaccinated	190 (90.9)	79 (90.8)	269 (100.0)
	Unvaccinated	19 (9.1)	8 (9.2)	27 (100.0)
**Vaccination time**	Before COVID-19	178 (85.2)	35 (40.2)	213 (100.0)
	After COVID-19	11 (5.3)	45 (51.7)	56 (100.0)
**Vaccination regimen**	Killed-virus	42 (72.4)	16 (27.6)	58 (100.0)
	mRNA	58 (59.8)	39 (40.2)	97 (100.0)
	Vector-based	29 (80.6)	7 (19.4)	36 (100.0)
**SARS-CoV-2 variant**	B.1.617.2 (Delta)	98 (79.0)	26 (21.0)	124 (100.0)
	BA.1 (Omicron)	101 (98.1)	2 (1.9)	103 (100.0)
	B.1.1.7 (Alpha)	7 (15.2)	39 (87.8)	46 (100.0)
	Wild Type	3 (13)	20 (87.0)	23 (100.0)
**COVID-19 symptoms at PCR testing**	Present	181 (86.6)	73 (83.9)	254 (100.0)
	Absent	28 (13.4)	14 (16.1)	42 (100.0)
**Hospitalization**	Yes	15 (7.2)	14 (16.1)	29 (100.0)
	No	194 (92.8)	73 (83.9)	267 (100.0)
**ICU admission**	Yes	4 (26.7)	4 (28.6)	8 (100.0)
	No	11 (73.3)	10 (71.4)	21 (100.0)

**Table 2 jcm-12-02613-t002:** The most commonly reported LC symptoms experienced by volunteers.

LC Symptoms	Gr1 n (%)	Gr2 n (%)	Total n (%)
**Fatigue**	34 (37.7)	22 (47.8)	56 (41.1)
**Cough**	27 (30.0)	8 (17.4)	35 (25.7)
**Memory problems**	22 (24.4)	6 (13.0)	28 (20.6)
**Dyspnea**	16 (17.8)	10 (21.7)	26 (19.1)
**Headache**	15 (16.7)	7 (15.2)	22 (16.2)

**Table 3 jcm-12-02613-t003:** Risk factors associated with OSC syndrome in Gr1. Abbreviations: OR, odds ratio; CI, confidence interval; ref, reference value. Significant *p* values were indicated with bold to assist the readers.

Risk Factors	OSC Symptoms		χ^2^ Test		Logistic Regression	
	Present n/N (%)	Absent n/N (%)	OR (95% CI)	*p* Value	OR (95% CI)	*p* Value
**Gender**						
Male	24/107 (22.4)	83/107 (77.6)	1 (ref)	**0.006**	-	-
Female	41/102 (40.2)	61/102 (59.8)	2.32 (1.27–4.25)			
**Age**						
12–17 years	2/18 (11.1)	16/18 (88.9)	-	0.143	0.31 (0.05–1.80)	0.192
18–55 years	57/170 (33.5)	113/170 (66.5)			1.26 (0.46–3.42)	0.649
≥56 years	6/21 (28.6)	15/21 (71.4)			1 (ref)	
**Smoking Status**						
Smoker	12/47 (25.5)	35/47 (74.5)	-	0.441	1.14 (0.27–4.86)	0.856
Non-smoker	50/147 (34.0)	97/147 (66.0)			1.72 (0.45–6.53)	0.427
Former-smoker	3/13 (23.1)	10/13 (76.9)			1 (ref)	
**Chronic disease**						
Present	25/55 (45.5)	30/55 (54.5)	2.38 (1.25–4.50)	**0.007**	-	-
Absent	40/154 (26.0)	114/154 (74.0)	1 (ref)			
**Vaccination status**						
Vaccinated	60/190 (31.6)	130/190 (68.4)	1.29 (0.45–3.76)	0.637		
Unvaccinated	5/19 (26.3)	14/19 (73.7)	1 (ref)			
**Vaccination Time**						
Before COVID-19	57/178 (32.0)	121/178 (68.0)	1.26 (0.32–4.90)	1.000		
After COVID-19	3/11 (27.3)	8/11 (72.7)	1 (ref)			
**COVID-19 symptoms at PCR testing**						
Present	64/181 (35.4)	117/181 (64.6)	14.77 (1.96–111.23)	**0.001**	-	-
Absent	1/28 (3.6)	27/28 (96.4)	1 (ref)			
**Hospitalization**						
Present	8/15 (53.3)	7/15 (46.7)	2.75 (0.95–7.94)	0.079		
Absent	57/194 (29.4)	137/194 (70.6)	1 (ref)			

**Table 4 jcm-12-02613-t004:** Risk factors associated with PSC syndrome in Gr2. Abbreviations: OR, odds ratio; CI, confidence interval; ref, reference value; *, chi square statistics could not be calculated. Significant *p* values were indicated with bold to assist the readers.

Risk Factors	PSC Symptoms		χ^2^ Test		Logistic Regression	
	Present n/N (%)	Absent n/N (%)	OR (95% CI)	*p* Value	OR (95% CI)	*p* Value
**Gender**						
Male	15/42 (35.7)	27/42 (64.3)	1 (ref)	0.988	-	-
Female	16/45 (35.6)	29/45 (64.4)	0.99 (0.41–2.39)			
**Age**						
12–17 years	2/5 (40.0)	3/5 (60.0)	-	*	1.87 (0.24–14.65)	0.553
18–55 years	24/63 (38.1)	39/63 (61.9)			1.72 (0.55–5.39)	0.350
≥56 years	5/19 (26.3)	14/19 (73.7)			1 (ref)	
**Smoking Status**						
Smoker	10/20 (50.0)	10/20 (50.0)	-	*	1.80 (0.57–5.67)	0.389
Non-smoker	21/62 (33.9)	41/62 (66.1)			1.71 (0.50–5.81)	0.315
Former-smoker	0/5 (0.0)	5/5 (100.0)			1 (ref)	
**Chronic disease**						
Present	9/26 (34.6)	17/26 (65.4)	0.94 (0.36–2.46)	0.897	-	-
Absent	22/61 (36.1)	39/61 (63.9)	1 (ref)			
**Vaccination status**						
Vaccinated	29/79 (36.7)	50/79 (63.3)	1.74 (0.33–9.17)	0.706		
Unvaccinated	2/8 (25.0)	6/8 (75.0)	1 (ref)			
**Vaccination Time**						
Before COVID-19	13/35 (37.1)	22/35 (62.9)	1.08 (0.43–2.68)	0.884		
After COVID-19	16/45 (35.6)	29/45 (64.4)	1 (ref)			
**COVID-19 symptoms at PCR testing**						
Present	30/73 (41.1)	43/73 (58.9)	9.07 (1.13–73.09)	**0.015**	-	-
Absent	1/14 (7.1)	13/14 (92.9)	1 (ref)			
**Hospitalization**						
Present	6/14 (42.9)	8/14 (57.1)	1.44 (0.45–4.61)	0.555		
Absent	25/73 (34.2)	48/73 (65.8)	1 (ref)			

**Table 5 jcm-12-02613-t005:** Association of LC symptoms with vaccination regimen and SARS-CoV-2 variants.

		LC Symptoms		
		Present n (%)	Absent n (%)	*p* Value
**Vaccination regimen**	Killed-virus	22 (37.9)	36 (62.1)	*0.159*
	mRNA	33 (34.0)	64 (66.0)	
	Vector-based	7 (19.4)	29 (80.6)	
**SARS-CoV-2 variant**	B.1.617.2 (Delta)	52 (41.9)	72 (58.1)	*0.395*
	BA.1 (Omicron)	49 (47.6)	54 (52.4)	
	B.1.1.7 (Alpha)	21 (45.7)	25 (54.3)	
	Wild Type	*14 (60.9)*	*9 (39.1)*	

## Data Availability

Data used for this study is available upon request from corresponding author.

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
