# Peer review of "Long-COVID-19 in Asymptomatic, Non-Hospitalized, and Hospitalized Populations: A Cross-Sectional Study"

_jcm, 2023, doi:10.3390/jcm12072613_

Round 1
Reviewer 1 Report
The manuscript of Aysegul Bostanci and colleagues investigated the prevalence of Long-COVID and determine the associated risk factors in Northern Cyprus. They found that initial viral cycle threshold (Ct) values of the SARS-CoV-2; vaccination status and type prior COVID-19; and SARS-CoV-2 variant displayed correlation in neither of the study groups. In addition, they found that persistent COVID-19 related symptoms can be detected after asymptomatic infection, and is not influenced by the vaccination status and SARS-CoV-2 variants. This study revealed some information; however, the significance is sufficient. In addition, the data analysis did not consider population bias, which would lead to inaccurate conclusions.
Major comments:
1. The participants are not evenly distributed. Are these factors being considered when the authors analyzed the data? For example: the vaccination status of the majority of these participants were vaccinated (above 90% in both group1 and 2). SARS-CoV-2 variant also has a large variation between group 1 and 2. Will these variation in the participants affect the results of this manuscript?
2. How do the authors define Health status by pre-covid of these participants?
Reviewer 2 Report
This paper investigates the occurrence of long COVID-19 syndrome in a population and presents clinical factors associated with the syndrome. This paper contributes knowledge to a handful of studies investigating long COVID-19 and would provide more insight into this rare phenomenon. However, the diagnosis of long COVID-19 is not confirmed by a health professional and was simply inferred through survey. Please see below the changes that will improve this paper:
1. Please explain how the prior diagnosis of COVID-19 was confirmed by the researchers, and how did the researchers identify the variants.
2. If the aim is to identify risk factors, please present relative risk with 95% confidence intervals in a table along with the corresponding P values.
3. Please explain "brain fog" and provide a definition. Were the long-COVID-19 symptoms entered in as a free text in the online survey or were there choices that the respondents can simply select?
4. Please enumerate the limitations and possible biases in this study given the study methodology.
5. How did the authors come up with the vaccine combinations in Table 5? On the same table, how were the variants identified?
6. Please explain why the authors said that some of the clinical categories cannot be analyzed due to small sample size. Please read about Fisher's exact and other post-hoc corrections for limited sample sizes.
Minor comments:
1. Please be consistent with the decimal places used in the paper. One decimal place is usually preferred, but p values should still be mentioned using three decimal places.
2. Figure 1 and Table 2 are redundant. Please choose one to present the symptoms.
Round 2
Reviewer 1 Report
The authors address my concerns.
Reviewer 2 Report
This version has greatly improved from the original draft. The tables and results are clearly presented. The only major comment is to provide explanation regarding the COVID-19 variants and how they were tested. Moderate English editing is also needed to finalize the paper.
